# YOLO-SASE: An Improved YOLO Algorithm for the Small Targets Detection in Complex Backgrounds

**DOI:** 10.3390/s22124600

**Published:** 2022-06-18

**Authors:** Xiao Zhou, Lang Jiang, Caixia Hu, Shuai Lei, Tingting Zhang, Xingang Mou

**Affiliations:** 1School of Mechanical and Electronic Engineering, Wuhan University of Technology, Wuhan 430070, China; zhouxiao@whut.edu.cn (X.Z.); 291834@whut.edu.cn (L.J.); shuailei@whut.edu.cn (S.L.); effyzh@hotmail.com (T.Z.); 2Beijing Aerospace Automatic Control Institute, Beijing 100000, China; whut_hucaixai@163.com

**Keywords:** infrared small target detection, super-resolution reconstruction, adaptive channel attention

## Abstract

To improve the detection ability of infrared small targets in complex backgrounds, an improved detection algorithm YOLO-SASE is proposed in this paper. The algorithm is based on the YOLO detection framework and SRGAN network, taking super-resolution reconstructed images as input, combined with the SASE module, SPP module, and multi-level receptive field structure while adjusting the number of detection output layers through exploring feature weight to improve feature utilization efficiency. Compared with the original model, the accuracy and recall rate of the algorithm proposed in this paper were improved by 2% and 3%, respectively, in the experiment, and the stability of the results was significantly improved in the training process.

## 1. Introduction

Compared with visible light, infrared imaging is less affected by fog, smoke, and other atmospheric obstacles and provides clear images in all weather conditions. It has been widely used in both civil and military fields [1]. For the small objects, due to the long distance between the targets and the detector, the imaging size is extremely small, occupying only dozens or even several pixels on the imaging plane. It also lacks detailed information such as color and texture, which increases the difficulty of target detection.

The task of the detection algorithm is to distinguish the target from the background and noise by extracting various features of the image, such as spatial domain features, frequency domain features, time-domain features, etc. Spatial and frequency domain features can be extracted from a single frame, while time-domain features must be extracted from multiple frames. At present, single-frame methods commonly used in infrared detection of small targets include filtering, local contrast, and frequency-domain transformation. When the background is complex or the target SNR (Signal Noise Ratio) is low, they often lead to more false alarms and require more parameter adjustment.

Deep learning methods have achieved good results in the field of target detection in terms of detection speed and accuracy. At present, object detection networks represented by Faster R-CNN [2], YOLO [3], SSD [4], etc., have shown fine performance with various datasets. However, the detection effect of infrared weak and small targets in complex backgrounds is not ideal in general. It is necessary to improve the characteristics of small targets with few features and little information. In the field of infrared small target detection, a batch of research using deep learning methods has appeared. Ju et al. [5] proposed an image filtering module is proposed to obtain the confidence map, aiming to enhance the response of infrared small targets and suppress the response of the background. Huang [6] et al. used multiple well-designed local similarity pyramid modules to improve the capture ability of infrared small target multi-scale features. Zhang et al. [7] proposed a method to generate synthetic TIR data from RGB data in infrared tracking. Wu et al. [8] transformed the detection of small targets into the classification of their distribution and proposed a classification network structure of a fully convolutional network domain. Ding et al. [9] proposed a multi-scale deformable convolution SSD network structure by combining pipeline filtering, and it performs better in multi-frame detection. Zhang et al. [10] used the traditional image preprocessing method to make the infrared image closer to the RGB image. To further improve the detection accuracy on the mainstream target detection network but makes training more difficult.

To further improve the detection precision and recall of tiny targets, we propose a multi-receptive field adaptive channel attention network based on the YOLO network and image super-resolution reconstruction method. The main contributions of this paper are:(1)The super-resolution model is used to reconstruct the fuzzy image, and the reconstructed image is input into the network for detection. The experiment verifies that the super-resolution reconstruction can improve the accuracy of target recognition.(2)A novel Self-Adaption Squeeze-and-Excitation (SASE) module is proposed. The SASE module can adaptively adjust the importance of different data channels in the training process and improve the stability of the model.(3)Drawing on the idea and method of multistage mixed dilatation convolution, the high-level features and low-level features of the image are fused by adding the RFB (Receptive field block) module, which increases the Receptive field and the expression ability of the feature image.

## 2. Proposed Approach

In the mainstream target detection network, the repeated downsampling in the convolutional layer is easy to cause the feature loss of small targets. We use the super-resolution method to upsampling a small target with a few pixels. By increasing the number of pixels of the small target before entering the convolutional layer, it can perform better in feature extraction and target detection. The target detection process is shown in Figure 1.

### 2.1. Infrared Image Super-Resolution Reconstruction Method Based on Generative Adversarial Network

The main information of infrared small targets in the complex background is high-frequency information. The signal of small targets can be retained in the super-resolution reconstruction process by using the generative adversarial network, which improves the detection precession of the model for small targets.

SRGAN (Super-Resolution Generative Adversarial Network) adds a discriminator network to the residual network-based image super-resolution reconstruction method. Its advantages are as follows: The mean square loss function (MSE) is replaced by antagonism loss and content loss. A generative adversarial network is introduced to replace the content loss of traditional pixel space with adversarial similarity. A deep residual network is introduced to extract richer image details.

The structure of the Generator Network and Discriminator Network of SRGAN is shown in Figure 2. The generator includes multiple residual modules, as well as a Batch Normalization (BN) and activation function layers. The discriminator contains eight convolution layers, activated by the Leaky ReLU function and finally connected to the full connection layer. SRGAN network uses perceptual loss and adversarial loss to achieve super-resolution reconstruction, which increases the clarity and fidelity of the restored image.

### 2.2. Infrared Small Target Detection Network

To detect infrared tiny targets quickly and accurately in a complex environment, multi-layer feature fusion is needed to improve the feature expression ability of the network for targets of different scales. After comparing the speed and accuracy of target detection algorithms based on regression, this paper adopts the YOLOv5 algorithm as the basic network of infrared small target detection. Darknet53 is used as the backbone network for feature extraction, and its construction method draws on the residual structure of Resnet to further enhance feature extraction capability [11]. The SASE module (self-adaption squeeze and congestion) is added after the backbone network and just before the detection layer. This improves the channel concentration of the network and makes network training more stable. A lower dimension YOLO detection scale is added to the detection output layer, which makes the detection layer correspond to more pixel values. Multi-layer feature fusion and multi-scale target detection are carried out by SPP(Spatial Pyramid Pooling) structure. A multi-receptive field module RFB (Receptive field block) composed of mixed expansion convolution is added to the lower two detection scales. The network structure is shown in Figure 3.

To increase the amount of information expressed by infrared image features and further improve the ability of the network to express features, an SPP module, as shown in Figure 4, was added to the network’s high-level semantic image detection layer. With the SPP module, the feature images used for target detection are fused with local features and global features, enriching the expression ability of feature images. The process is as follows: First, the input features are convolved to halve the number of channels. Then, the maxpooling of different sizes is carried out. The core size of pooling is 5, 9, and 13, respectively, and the step size is 1. Finally, the three pooling layers are combined with the original feature graph after downsampling and input to the next convolutional layer for feature learning.

The high dimension feature graph information is obtained by the SASE modules at each detection layer branch behind the feature extraction backbone network. The structure of the SASE module is shown in Figure 5. It is based on the SEnet (squeeze and congestion networks) structure widely used in residual [12] and inception networks [13]. In the process of network training, the feature graph with a greater detection function is given a higher weight, thus speeding up the network training and improving the ability of the whole network to select and capture features. In the SASE module, the similarity measure of channel weight transformation is added. The Pearson correlation coefficient is used to measure the similarity of the two tensors $x_{1}$ and $x_{2}$. For the transformation less than the similarity threshold α, tensor addition of residual structure is adopted. For the transformation greater than the similarity threshold α, direct replacement is adopted.
(1)$$f=\left\{\begin{array}{r} \left(x_{1}+x_{2}\right)/2,\;\;f_{1}\left(x_{1},x_{2}\right)<\alpha \\ x_{2},\;\;f_{1}\left(x_{1},x_{2}\right)\geq \alpha \end{array}\right.$$
where $x_{1}$ is the input tensor, $x_{2}$ is the output tensor, $f_{1}$ is the Pearson correlation coefficient for calculating the weight of channel C in the two tensors, α is the similarity threshold.

As shown in Figure 6, the RFB module [14] is added to the two low-level detection output layers, referring to the idea of integrating high-level semantic features with low-level detail features of the feature pyramid. High-level semantic information is added to the low-level detection output layer that pays attention to detailed information, thus strengthening the feature expression ability of the network for multi-scale targets. The RFB module is to connect 1 × 1, 3 × 3, and a 5 × 5 convolution kernel, respectively, through two 1 × 1 convolution layers and combine three 3 × 3 convolution kernels with expansion rates of 1, 3, and 5, respectively, to achieve the effect of covering a larger area through concatenate splicing, while the central area has a larger weight.

## 3. Experiment and Analysis

To verify the ability of the detection network proposed in this paper to detect small infrared targets in a complex background, experiments were carried out on infrared target datasets in different backgrounds. Experiments were designed to verify the effectiveness of the image reconstruction methods and network modules. The detection results were compared with existing typical network models.

### 3.1. Experimental Details

The details of the relevant parameters of the experiment are shown in Table 1 below. The training process is accelerated by GPU.

The dataset covers the sky, ground, and other backgrounds as well as a variety of scenes, totaling 22 pieces of data, 30 tracks, 16,177 frames of images, and 16,944 targets. Considering that the total number of images is large and the time and hardware requirements for network training are high, this paper evenly selected one-fifth of the images as the experiment dataset and divides it into training (90%) and test sets (10%). Part of the infrared small target dataset is shown in Figure 7.

### 3.2. Contrast Test of Reconstruction Image

To evaluate the effect of image reconstruction, Peak Signal to Noise Ratio (PSNR), Structural Similarity (SSIM), precision, and recall rate of the target detection network are used. Figure 8 shows the effect of bilinear interpolation, nearest-neighbor interpolation, and SRGAN network for the quadruple upsampling reconstruction of infrared small targets.

PSNR is inversely proportional to the logarithm of mean square error (MSE) of the SR image, as shown in Formula (2). PSNR can reflect the noise level of the image and represent the distortion of the reconstructed image. The larger the value is, the better the reconstructed image will be.
(2)$$\mathrm{PSNR}=10\times \log\frac{W\times H\;{\left(2^{n}-1\right)}^{2}}{\sum_{\begin{array}{c} 0<x\leq \;W\\ 0<y\leq \;H\; \end{array}}{\left[I_{HR}\left(x,y\right)-I_{SR}\left(x,y\right)\right]}^{2}}$$
where *H* and *W* are the height and width of the image, respectively, (x, y) are the coordinates of each pixel point. 

SSIM calculated by Formula (3) reflects the similarity of brightness, contrast, and structure between generated image SR and high-resolution image HR.
(3)$$\mathrm{SSIM}=\frac{\left(2\mu_{HR}\mu_{SR}+C_{1}\right)\left(2\sigma_{HR,SR}+C_{2}\right)}{\left(\mu_{HR}^{2}+\mu_{SR}^{2}+C_{1}\right)\left(\sigma_{HR}^{2}+\sigma_{SR}^{2}+C_{2}\right)}$$
where μ represents the gray mean value, σ represents variance, $C_{1}$ and $C_{2}$ are constants that keep the equation valid.

The evaluation indexes were calculated for the original images and reconstructed images by different methods shown in Table 2. Compared with Bilinear and Nearest, SRGAN had better performance in PSNR and SSIM, which means that the reconstructed image by SRGAN is closer to the original image. At the same time, the SRGAN method has a better value than the original image in precision and recall rate.

Compared with the bilinear and nearest, the SRGAN method takes more time to calculate, the computational cost is 0.92 s for each image in super-resolution by SRGAN. However, the SRGAN method has a better effect on image reconstruction and subsequent target detection. The high time consumption is caused by the SRGAN method itself, which is expected to be solved in the follow-up research of super-resolution methods. Moreover, higher PSNR and SSIM results in higher confidence scores and detection performance in the target detection network.

According to the statistics in Figure 9, the *x*-axis represents the target confidence scores of different segments from 0 to 1, and the *y*-axis represents the number of images placed in this segment. Compared with the original image, the confidence score of the reconstruction image with the super-resolution has significantly improved. The number of undetected images with confidence less than 0.1 has significantly decreased, and the number of images with a different confidence level higher than 0.1 has increased.

The experimental results are shown in Figure 10. As can be seen from Figure 9, the confidence score of the reconstructed images is higher than the original images, which means it made small targets easier to be detected in images.

### 3.3. Contrast Test of Network Structure

To evaluate the detection effect of infrared small targets, average precision (AP), average recall (AR), and frame per second (FPS) were used as evaluation indexes. To verify the effectiveness and reliability of the proposed detection algorithm, the experiment was divided into three parts: overall effect comparison, module improvement comparison, and ablation experiment. Original infrared images were used in the training set. Figure 11a shows that, compared with the origin YOLO algorithm, the accuracy of YOLO_SASE proposed in this paper was significantly improved after training 50 epochs and finally achieved a higher accuracy level. In Figure 11b of recall rate, the red line is the original data of YOLO_SASE, the blue line is the original data of YOLO_origin, the green line is the smoothed YOLO_SASE data, and the purple line is the smoothed YOLO origin data. It can be seen that the recall rate of the YOLO_SASE is basically higher than that of the original method; that is, the improved algorithm YOLO_SASE is superior to the original algorithm in overall effect.

To verify the improvement of network model training and detection performance by the SASE module, the original YOLO network, and the standard SE module and SASE module (α selected 0.7, 0.3, and 0) were used for the contrast test. Weight files are all pre-weights trained by MS COCO datasets. Experimental results of detection precision and recall rate are shown in Table 3. The standard deviation of precision and recall rate during the training process in the grouped experiment is shown in Figure 12. Compared with the original YOLO and standard SE modules, the SASE module improves the stability of accuracy and recall rate during training.

The algorithm models for comparison were divided into 11 groups in the experiment, and the experimental results are shown in Table 4. The first group is the original YOLO algorithm; The second group added the standard SE module; Groups 3, 4, and 5 were SASE modules with different thresholds. The sixth group added an SPP module; Group 7, 8, 9, and 10 were added SE and SASE modules with different thresholds based on the SPP module, and group 11 adjusted the number of output characteristic layers and added an RFB module based on the previous group.

Compared with the original model in group 1, the precision and recall rate of the algorithm proposed in this paper in group 11 were improved by 2% and 3%

To further verify the detection performance of the proposed algorithm, we compared it with Faster R-CNN [2], RetinaNet [16], SSD [4], and YOLOv3 [11]. The test results are shown in Table 5.

It can be seen that the algorithm proposed in this paper achieves the best accuracy and recall rate under the condition of small speed loss.

## 4. Discussion

In this paper, the YOLO_SASE algorithm can improve the channel attention of the target detection network. Compared with other networks and the original network, the presented has better precision and recall rate. However, the calculation of super-resolution reconstruction using SRgan is time-consuming and cannot achieve real-time detection. In the next step, the structure of the infrared small target itself will be analyzed, and the network parameters will be compressed to further improve the detection efficiency of the infrared small target.

In the infrared small target dataset adopted in this paper, part of the sequence data itself has low target SCR (signal-to-cluster ratio) and is difficult to detect. These difficult parts limit the further improvement of the target detection algorithm, and it is difficult to solve the problem only through the single frame target detection method. We listed the sequence of low SCR of partial target imaging in the original dataset, and conducted experiments on these images with different commonly used target detection networks. The experimental results are as follows in Table 6:

In the process of uniformly selecting all samples of the dataset, it is inevitable to encounter several very difficult samples in Figure 13. These low SCR samples accounted for 22.19% of the overall dataset.

The first type: airplane target pixel is too few in Figure 14. For example, in the Data21 sequence image, the target occupies only two pixels in most cases.

The second type: target and background building highlight part fusion in Figure 15. When the target flies over backgrounds of varying grayscale intensities, some of the highlighted backgrounds blend in with the highlighted target.

The third type: target and background building dark part fusion in Figure 16. When the target flies over a background of different grayscale intensities, some of the low brightness background merges with the low brightness part of the target.

The fourth type: picture shooting drag shadow. In the original dataset, the shooting target is in constant motion, and the camera shooting will move intermittently. In the process of moving can produce a shooting drag; at this time, the small target is blurred and may even disappear. For example, in the lower right corner of Figure 13.

Table 7 below shows the experimental results of Low SCR samples in Table 6 above. It can be seen that for this part of the samples, several target detection networks cannot reach a precision of more than 70%. Note some Low SCR images in the dataset are not suitable for single-frame target detection, and these parts limit the further improvement of the target detection algorithm.

## 5. Conclusions

To solve the problem of infrared small target detection in complex background, this paper designed a deep learning target detection algorithm based on the YOLO algorithm with an image super-resolution reconstruction method. Based on the standard SE module, an adaptive channel attention SASE module was proposed. The ability of shallow feature extraction is improved by adjusting the number of detection output layers and adding a multi-field fusion RFB module. Using small targets in the infrared image sequences data as the experimental dataset, the results show that the proposed algorithm based on the original model can effectively improve the accuracy and recall rate, stability, and increase the training process.

## Figures and Tables

**Figure 1 sensors-22-04600-f001:**
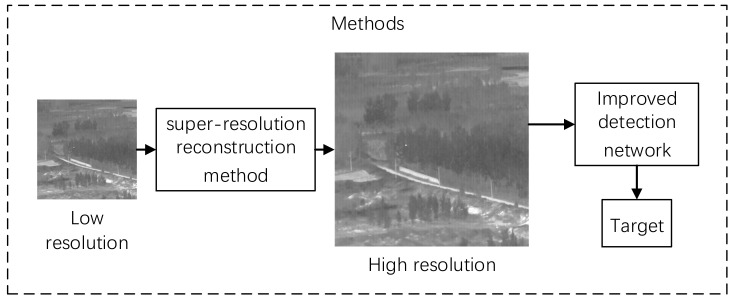
Target detection process.

**Figure 2 sensors-22-04600-f002:**
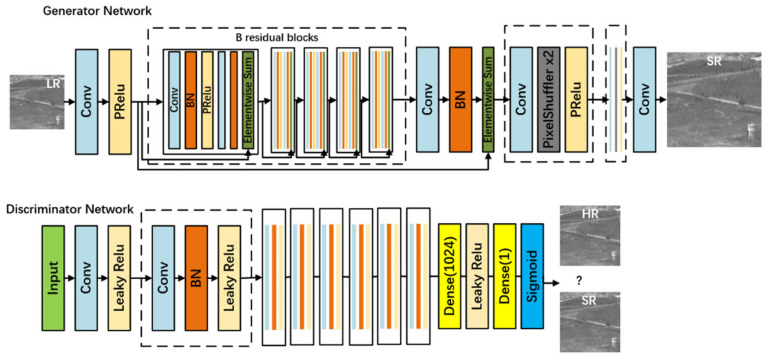
SRGAN generator G and discriminator D model.

**Figure 3 sensors-22-04600-f003:**
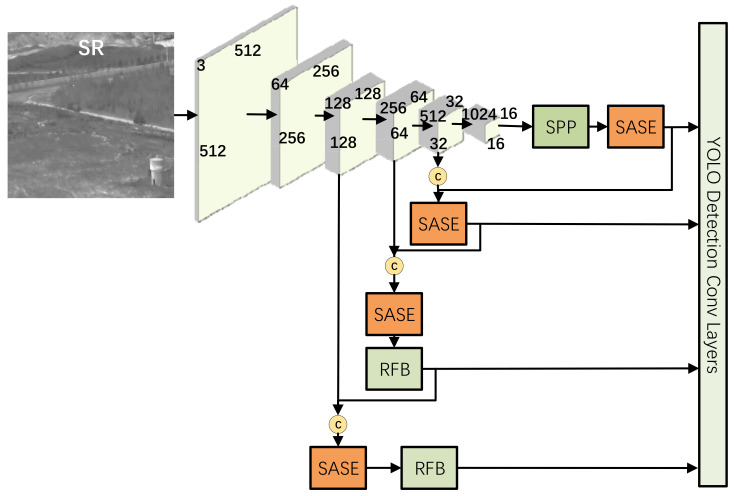
Network structure diagram.

**Figure 4 sensors-22-04600-f004:**
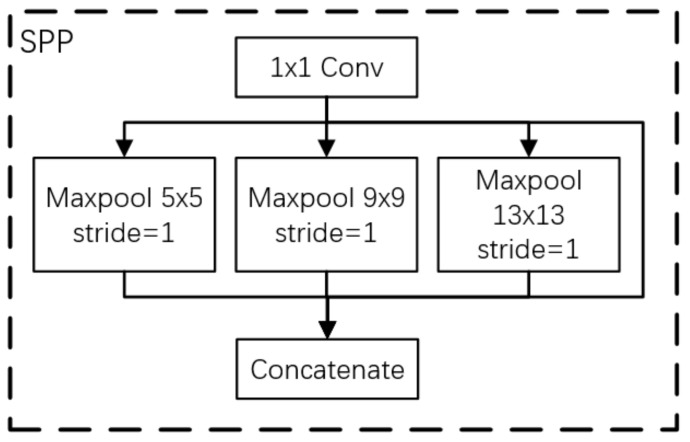
SPP module.

**Figure 5 sensors-22-04600-f005:**
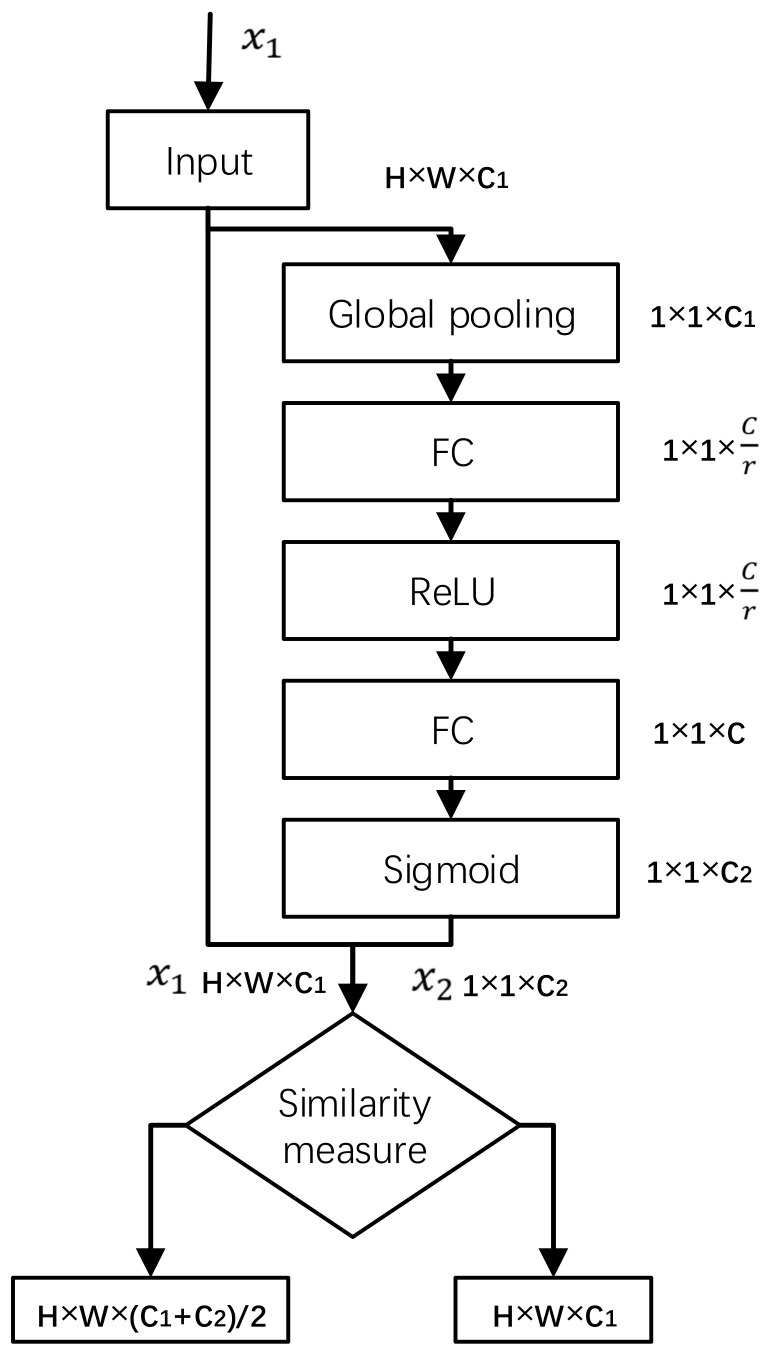
SASE module.

**Figure 6 sensors-22-04600-f006:**
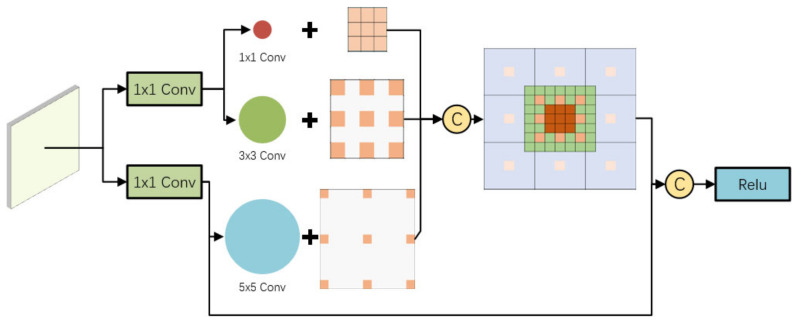
RFB module.

**Figure 7 sensors-22-04600-f007:**
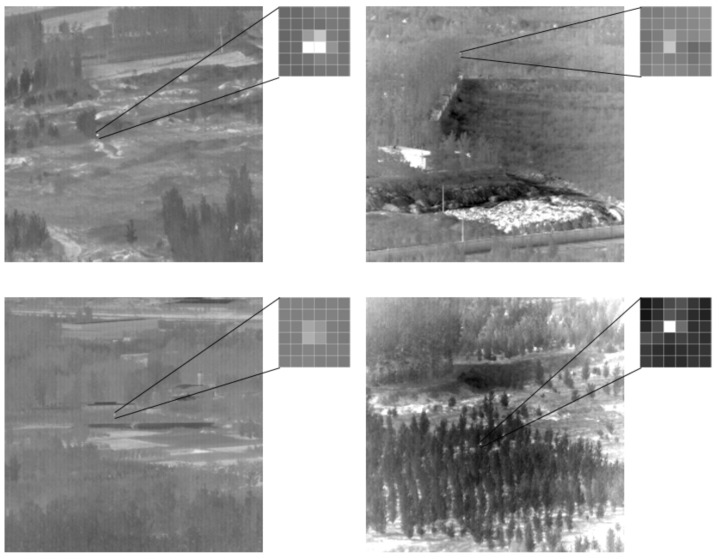
Infrared small target dataset.

**Figure 8 sensors-22-04600-f008:**
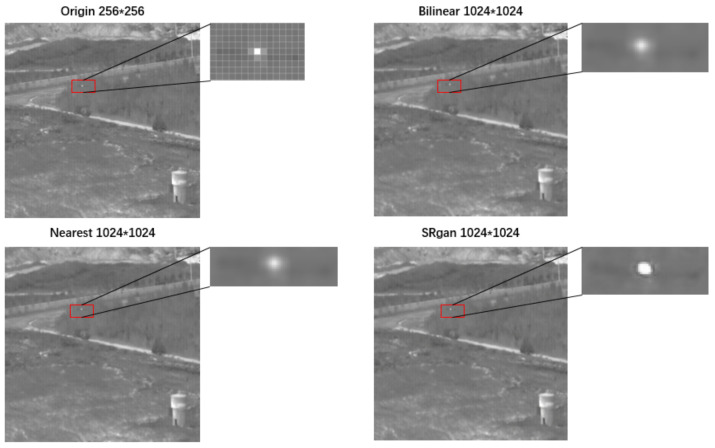
Image reconstruction of infrared small target by quadruple upsampling.

**Figure 9 sensors-22-04600-f009:**
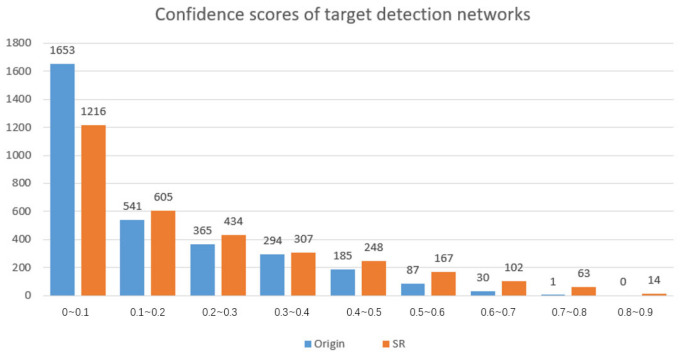
The confidence score statistical histogram of Origin and SR images in target recognition network detection.

**Figure 10 sensors-22-04600-f010:**
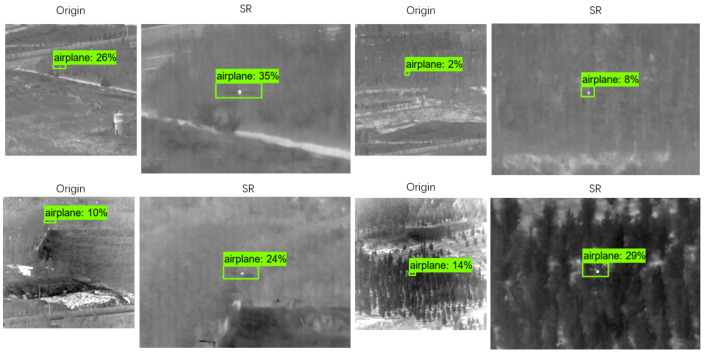
Example of confidence degree of the partial original image and reconstructed image in target recognition network detection.

**Figure 11 sensors-22-04600-f011:**
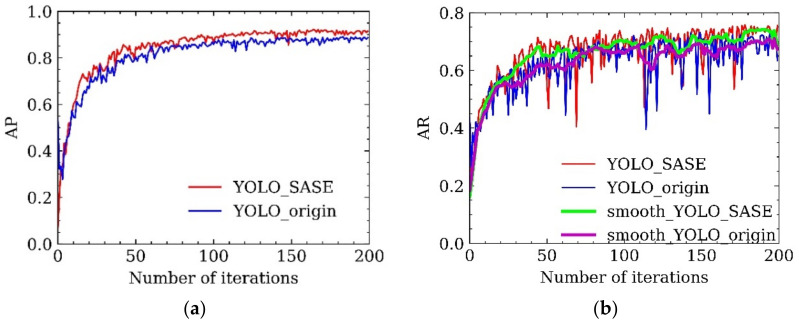
(**a**) accuracy comparison; (**b**) recall comparison.

**Figure 12 sensors-22-04600-f012:**
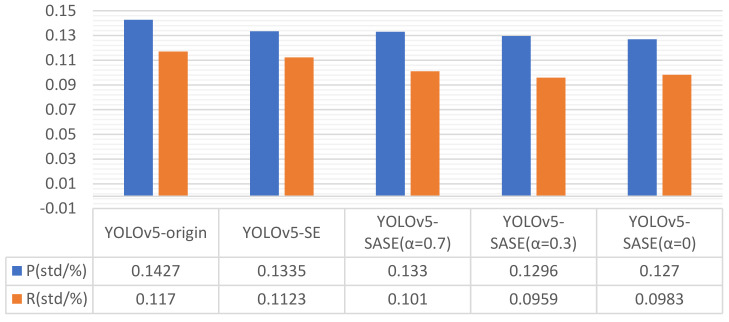
Stability comparison experiment of SASE module training results. Standard deviation is taken from the last 50 data in 200 epochs.

**Figure 13 sensors-22-04600-f013:**
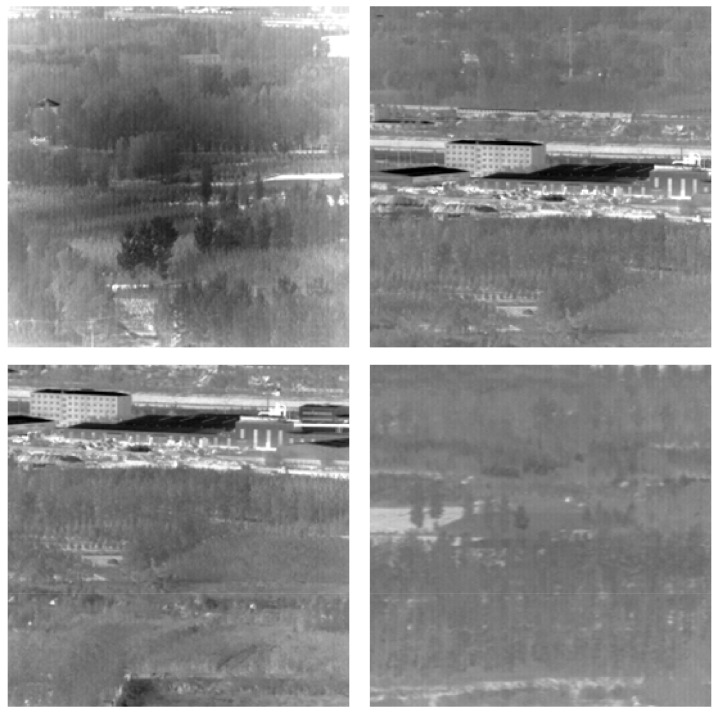
Four kinds of hard samples.

**Figure 14 sensors-22-04600-f014:**
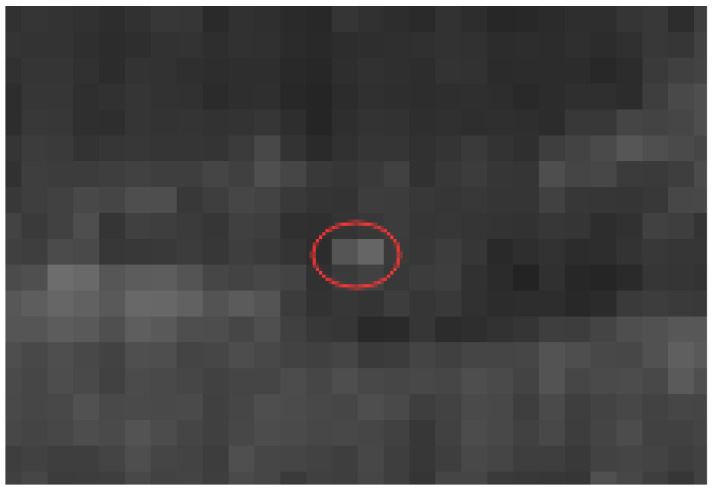
Few pixels target.

**Figure 15 sensors-22-04600-f015:**
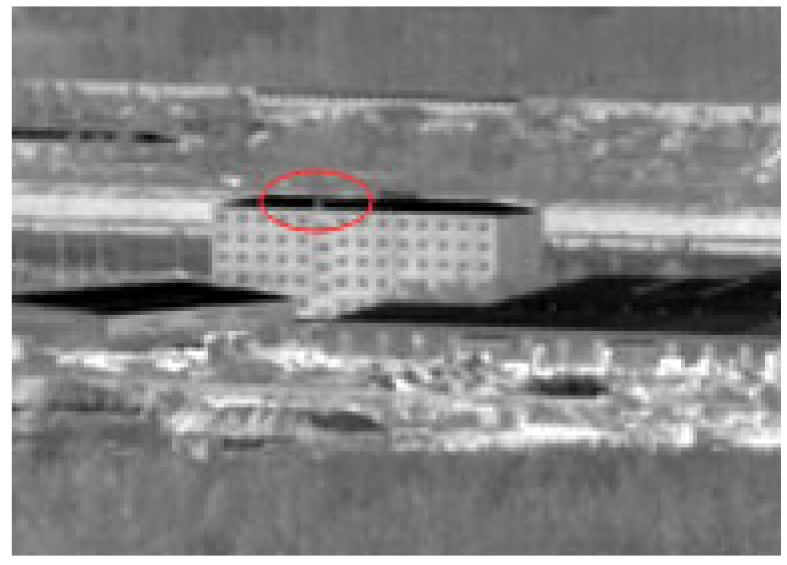
Target fusion with highlight background.

**Figure 16 sensors-22-04600-f016:**
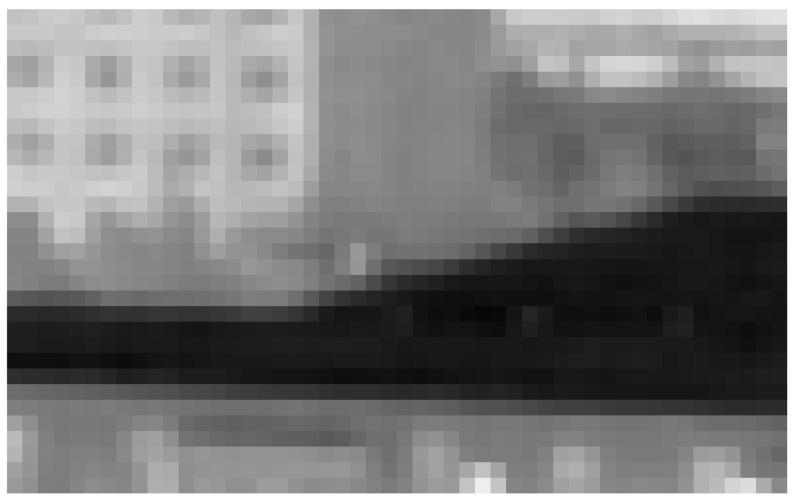
Target fusion with a dark background.

**Table 1 sensors-22-04600-t001:** Experimental environmental parameters.

	Parameters
Dataset	a dataset for infrared detection and tracking of dim-small aircraft targets underground/air background [15]
Environment	Ubuntu
Framework	Pytorch YOLO
CPU	Intel(R) Xeon(R) Gold 5218 CPU
GPU	Tesla T4
Memory	32 GB
Optimal	Momentum, 0.937
Learning rate	Initial learning rate 0.001, cosine function
Batch Size	32
Epoch	200

**Table 2 sensors-22-04600-t002:** PSNR results of image reconstruction and target recognition network detection results.

Methods	PSNR	SSIM	Precision/%	Recall/%
Original	-	-	89.07	67.53
Bilinear	30.06	0.8894	88.98	60.63
Nearest	32.31	0.8655	78.97	59.04
SRGAN	41.86	0.9836	90.03	68.08

**Table 3 sensors-22-04600-t003:** Comparison experiment of P/R effect of SASE module, training epochs were 200.

Model	Precision/%	Recall/%	FPS
YOLO_origin	87.90	68.7	85.47
YOLO_SE	88.13	67.31	83.21
YOLO_SASE (α = 0.7)	87.89	70.15	82.14
YOLO_SASE (α = 0.3)	88	68.47	81.96
YOLO_SASE (α = 0)	88.16	69.3	79.64

**Table 4 sensors-22-04600-t004:** Results of ablation experiment, training epoch were 200.

Group	SPP	SE	SASE (0.7)	SASE (0.3)	SASE (0)	RFB	P/%	R/%	FPS
1							87.90	68.7	85.47
2		✓					88.13	67.31	83.89
3			✓				87.89	70.15	83.04
4				✓			88	68.47	82.26
5					✓		88.16	69.3	80.94
6	✓						89.07	67.53	83.95
7	✓	✓					89.89	69.46	83.21
8	✓		✓				89.69	69.23	82.14
9	✓			✓			89.85	71.01	81.96
10	✓				✓		90.06	69.99	79.64
11	✓				✓	✓	90.22	71.34	72.99

**Table 5 sensors-22-04600-t005:** Results of different detection networks.

Method	P/%	R/%	FPS
Faster R-CNN	50.6	63.9	12.67
RetinaNet	79.33	61.23	97.6
SSD	88.29	53.92	69.43
YOLOv3	87.90	68.7	85.47
YOLO_SASE	90.22	71.34	72.99

**Table 6 sensors-22-04600-t006:** Low SCR sequence.

Sequence	Frame	SCR
Data10	401	0.38
Data13	763	1.98
Data14	1426	1.51
Data17	500	1.09
Data21	500	0.42

**Table 7 sensors-22-04600-t007:** Method results in Low SCR sample.

Method	P/%	R/%
Faster R-CNN	33.97	55.38
RetinaNet	49.67	51.98
SSD	61.71	47.51
YOLOv3	56.37	58.19
YOLO_SASE	68.95	61.73

## Data Availability

These data were derived from the following resources available in the public domain [15]: A dataset for infrared image dim-small aircraft target detection and tracking under ground/air background.

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
