# Peer review of "YOLO-SASE: An Improved YOLO Algorithm for the Small Targets Detection in Complex Backgrounds"

_sensors, 2022, doi:10.3390/s22124600_

Round 1
Reviewer 1 Report
In the article, the Authors proposed an improved deep learning target detection algorithm based on the YOLO algorithm with an image super‐resolution reconstruction method. They described the various stages of the method's operation. Compared with other methods and they proved the effectiveness of its operation.
The content of the article suggests that the super-resolution reconstruction method (increasing the resolution of the photo) plays an important role in the obtained results, not the YOLOv5 algorithm, as the title of the article suggests. In turn, the algorithms for increasing the resolution are known and widely described.
In Figure 9, in x axis the 0.3-0.4 range is omitted from the results, what was the reason?
In Figure 12, the y-axis scale starts at 0.09, which may be confusing to the reader as it suggests a significant difference between the P and R scores.
In line 253 there was no reference to YOLOv3, was it pos. 10 from the literature? The results of YOLOv3 were used in tab. 5.
The list of references requires a complete reconstruction, different styles are used in it, in pos. 3 the example text form hasn’t been removed. Literature review at the moment is only done for one country.
Author Response
We have a point-by-point response , a highlight and a clean versions.

Reviewer 2 Report
In this work, the authors present a novel neuronal network suitable for detection of infrared small targets in complex backgrounds. The algorithm is based on the YOLO detection framework and SRGAN network, taking super‐resolution reconstructed images as input, combined with SASE module, SPP module, and multi‐level receptive field structure, while adjusting the number of detection output layers, through exploring feature weight to improve feature utilization efficiency. Here some issues that encourage the authors to address:
1 - I can understand the idea and advantages regarding to the use of super resolution images however, more discussion about the computational costs of this operation is needed.
2 - For the results section I think that the improvement is insignificant and therefore, is not enough for justify the manuscript acceptance. I think that more discussion about the advantages of the proposed approach compare with the current literature could clarify this issue.
Author Response
We have three file ,one for point-by-point response ,one for highlight version,one for clean version.

Reviewer 3 Report
The presented work is based on YOLO algorithm for small targets in complex scenarios. This studt seems to be good and the proposed method and analyses are well supported. However, the Conclusion chapter is missing. Please include this section.
thank you
Author Response

(The authors gave the same response as above.)

Round 2
Reviewer 1 Report
The authors referred to the comments of the reviewer. Most of the suggestions were taken into account in the responses sent. However, in the final version of the manuscript, Fig. 9 (still an old drawing) and Fig. 12 do not coincide with the drawings presented in the cover letter. Also, the literature review was not done carefully.
The values on the horizontal axis in Fig. 9 (with the cover letter) at present raise even greater reservations. The authors changed the values without explaining anything. It follows that the values on the horizontal axis can be taken freely. Why is the range 03-04 missing?
Author Response
point by point response

This manuscript is a resubmission of an earlier submission. The following is a list of the peer review reports and author responses from that submission.